# Preparation and Testing of a Palladium-Decorated Nitrogen-Doped Carbon Foam Catalyst for the Hydrogenation of Benzophenone

**DOI:** 10.3390/ijms241512211

**Published:** 2023-07-30

**Authors:** Ádám Prekob, Viktória Hajdu, Zsolt Fejes, Ferenc Kristály, Béla Viskolcz, László Vanyorek

**Affiliations:** 1Institute of Chemistry, University of Miskolc, H-3515 Miskolc-Egyetemváros, Hungary; viktoria.hajdu@uni-miskolc.hu (V.H.); zsolt.fejes@uni-miskolc.hu (Z.F.); bela.viskolcz@uni-miskolc.hu (B.V.); 2Institute of Mineralogy and Geology, University of Miskolc, H-3515 Miskolc-Egyetemváros, Hungary; ferenc.kristaly@uni-miskolc.hu

**Keywords:** hydrogenation, catalyst development, palladium, carbon support, nitrogen doped, carbon foam

## Abstract

Catalytic activity of a palladium catalyst with a porous carbon support was prepared and tested for benzophenone hydrogenation. The selectivity and yields toward the two possible reaction products (benzhydrol and diphenylmethane) can be directed by the applied solvent. It was found that in isopropanol, the prepared support was selective towards diphenylmethane with high conversion (99% selectivity and 99% benzophenone conversion on 323 K after 240 min). This selectivity might be explained by the presence of the incorporated structural nitrogens in the support.

## 1. Introduction

Diphenylmethane (DPM) is a common organic compound in perfumery, cosmetics, dye, and coating industries [1,2]. Furthermore, it is a basic material for the production of AIE-active (aggregation-induced emission) luminogens. These luminogens (e.g., tetraphenylethene) have outstanding light-emitting properties and do not lose their emission efficiency due to the so-called aggregation-caused quenching effect; therefore, they are often used in the OLED (organic light-emitting diode) technology [3,4,5].

Diphenylmethane is mainly prepared through Friedel–Crafts reaction [6], but the catalytic hydrogenation of benzophenone can also be a suitable method [7]. During the hydrogenation, either diphenylmethanol (benzhydrol) or, as an over-hydrogenated product, diphenylmethane is produced. If the main goal is to prepare an over-hydrogenated product, highly efficient carbon-based catalysts can be ideal for use. According to the literature, many different catalysts have already been tested in benzophenone hydrogenation. Kumbhar et al. tested monometallic (Ni) and bimetallic (Ni-Cu and Ni-Fe) catalysts on SiO_2_ and TiO_2_ supports in benzophenone hydrogenation [8]. They also investigated the effect of different solvents like cyclohexane, isopropanol, methanol, and the mixture of methanol and water. Based on their results, all of the catalysts were selective to benzhydrol in every solvent, and this selectivity increased with the solvent polarity. The highest benzhydrol yield was achieved in a methanol–water solution in the presence of NaOH using a Ni-Fe/TiO_2_ catalyst (88% benzophenone conversion and 98.4% benzhydrol selectivity). Bai and his colleagues prepared a La-doped Ni-B alloy catalyst for benzophenone hydrogenation and measured 99.8% benzophenone conversion with 90.0% benzhydrol selectivity in methanol. Both values were higher than the commonly used Raney nickel (31.3% conversion and 78.1% selectivity) [9]. Bawane et al. tested commercial Pd/C, Pt/C, and Raney nickel catalysts in isopropanol [10]. According to their results, almost 100% benzhydrol selectivity was detected in all cases.

It has already been proven that the different noble-metal-decorated carbon forms are excellent catalysts in many types of catalytic reactions [11,12,13,14,15,16]. It is also well known that carbon supports with various special structures (e.g., carbon nanotubes and carbon foams) often provide more advantageous properties than the simpler ones, like carbon black or activated carbons [17].

Doping is an excellent way to modify basic carbon structures and therefore their properties [18,19,20]. The simplest way of doping is the so-called in situ doping, where usually a nitrogen (or other heteroatom)-containing carbon source is decomposed, and part of the heteroatom content covalently incorporates into the structure of the produced carbon form [21,22]. Nitrogen-doping results in the creation of different nitrogen-containing sites (e.g., pyridinic, pyrrolic, or graphitic), which show structural defects like vacancies where the catalytically active metal particles are more likely to be adsorbed with stronger interactions [23,24,25]. This phenomenon allows carbon-supported catalysts to provide a higher reaction rate, conversion, and product selectivity.

Foam-like catalyst supports are extensively researched materials due to their outstanding heat conductivity, which makes them ideal for highly exothermic reactions, e.g., hydrogenation [3]. Furthermore, foams provide a high external surface area, which makes them suitable for external surface-limited reactions. They also facilitate mass transport, which in turn increases the reaction rate [26]. Foam-like carbon products can be prepared, for example, by reacting 4-nitroaniline and sulfuric acid at an elevated temperature [27]. Since 4-nitroaniline has a high nitrogen content, it incorporates into the foam structure and exerts beneficial effects in catalysis.

In this paper, we present the preparation of a nitrogen-doped carbon foam catalyst, its characterization, and its test results in benzophenone hydrogenation.

## 2. Results and Discussion

### 2.1. Characterization of the CS Catalyst Support

The surface area (based on BET method) of the carbon foils was 3.36 ± 0.03 m^2^/g, which increased to 11.03 ± 0.05 m^2^/g after activation at 1173 K in CO_2_ atmosphere.

The CS was examined by using SEM. The membranous, foil-like structure of the carbon form can be seen in Figure 1 A,B. These nitrogen-doped carbon foils are very advantageous as supports for palladium nanoparticles during the catalyst preparation because on the surface of these layers, the nanoparticles can be easily anchored. Moreover, the catalytically active metal particles are located on the outer surface of the carbon nanofoils, which makes them easily accessible for the reactant molecules; thereby, the catalytic reaction is faster than in the case of a porous catalyst. Originating from the precursor, nitrogen atoms were incorporated into the carbon foils as pyridinic-, pyrrolic-, and graphitic-type nitrogens, which were identified by XPS technique. The deconvoluted N1s band of the spectrum showed four peaks identified at 398.0, 400.2, 401.4, and 404.6 eV binding energy, which are associated with pyridinic, pyrrolic, graphitic, and oxidized nitrogen, respectively (Figure 1C). The amount of nitrogen doped into the CS is 13.4 at% as calculated by XPS elemental analysis. The different incorporated nitrogen forms are as follows: 43.6% pyridinic, 15.0% graphitic, 9.2% pyrrolic, and 15.0% oxidized. These nitrogen atoms incorporated the CS structure have significant influence on the carbon structure. The bond lengths of pyridinic C-N (1.41Å) and graphitic C-C (1.42Å) are similar; however, the pyrrolic sp^3^ nitrogen disrupts the six-membered ring structure of the above-mentioned carbon forms. Another effect of the pyridinic nitrogens, namely that they led to formation of monovacancies, caused defects in the carbon structure. In this sense, occurrence of nitrogen atoms in the honeycomb-like lattice are accompanied with various structure defects (vacancies, bonding disorders, and non-cyclized structures) that in turn create high disorder in the structure. Moreover, due to the high electronegativity and smaller covalent radius of nitrogen, the doping would significantly influence not only the structural but the electronic properties of the CS. Thus, anchoring of the catalytically active metals (i.e., palladium) will be more favorable around the defects and the doping nitrogen atoms [28]. Furthermore, nitrogen doping of carbon can also influence the selectivity of the catalyst. In the case of heterogeneous hydrogenation reactions, for example, higher selectivity was reported towards aminobenzaldehyde by using a Pd/N-doped carbon nanotube catalyst in reduction of nitrobenzaldehyde, which was mainly attributed to the presence of Pd–N complexes. These surface Pd–N complexes decrease the sintering of the palladium nanoparticles, thus improving the stability and the dispersion of the Pd particles [29]. Enhanced selectivity was experienced towards aminobenzaldehyde with Pd-containing nitrogen-doped carbon nanotubes, which is explained by the promoting effect of NBCNTs supports.

To identify the surface functional groups on the carbon foils of CS, the C 1s band on the XPS spectrum was deconvoluted (Figure 1D). In the case of the CS, the intense peaks at 284.5 eV could be assigned to the sp^2^ carbon atoms of the graphitic regions. This demonstrates that most of the carbon atoms are arranged in a honeycomb lattice in the carbon layer. The deconvoluted C 1s band contains five weak peaks centered at 284.9, 285.5, 287.2, 287.8, and 289.2 eV binding energy, which correspond to the C-C or C-H, C-N, C-OH and C-O-C, C=O, C-N or C-O, C=O, and O=C-O groups, respectively, which are formed during the oxidation step (in CO_2_ atmosphere at 723 K) of the activation method. The total oxygen content of the CS is 5.9 at%. The presence of these oxygen-containing functional groups, namely -OH and -COOH, in the carbon support is very useful. These groups can be deprotonated and cause a negatively charged surface, which in turn contributes to the anchoring of the catalytically metal ions onto the catalyst support during preparation of the catalyst.

The catalyst support was further examined by FTIR measurements (Figure 2A). In the FTIR spectrum oxygen-containing functional groups were found at 1195 cm^−1^, 1701 cm^−1^, and 3432 cm^−1^, which belong to the stretching vibration modes of the C–O, C=O (of carboxylic structures), and -OH groups, respectively. The latter may indicate the presence of adsorbed water [30]. The bending vibration mode of the -OH groups resulted in a band at 1398 cm^−1^. Due to the incorporated structural nitrogen, the C–N, C=N, and -NH_2_ groups were also present, as indicated by the bands at 1458 cm^−1^, 1625 cm^−1^, and 3167 cm^−1^ wavenumbers, respectively [31]. The band at 1620 cm^−1^ can be assigned to the skeleton vibration mode of the carbon structure (νC=C bonds) and stretching vibration of the C=N bonds [32,33]. Moreover, the symmetric and asymmetric vibration of the -CH bonds are visible at 2855 cm^−1^ and 2929 cm^−1^, respectively. The presence of the oxidized nitrogen was confirmed by the νNO vibration band at 1472 cm^−1^. The band at 1122 cm^−1^ belongs to the vibrations of physically adsorbed SO_2_ [34]. More sulfur-containing functional groups were identified at 842 cm^−1^ (νS-O) and 1034 cm^−1^ (νS=O), which can be explained by the presence of sulfuric acid during the carbonization step of 4-nitroaniline. The sulfur content was also confirmed by XPS measurement (Figure 2B). The oxygen atom of the -OH group becomes electron rich after deprotonation in aqueous medium. This leads to an increased negative surface charge of the CS, indicated by the negative electrokinetic (Zeta) potential (−24.8 eV) measured in the aqueous medium (Figure 2C). The negative zeta potential is advantageous because the CS carrier is highly dispersible in aqueous medium. The polarity of the carbon surface contributes to adequate wetting. Ions of catalytically active metals bind to the surface of the carbon support due to the presence of dissociable functional groups (-OH groups). For the distribution of the catalytically active metal to be uniform on the surface of the support, it is necessary for the CS support to form a stable, homogeneous dispersion during the preparation of the catalyst. As a result, the entire surface of the support becomes easily accessible to the ions of the catalytically active metal (e.g., Pd and Pt), where they can be anchored by means of ion-exchange adsorption or surface complex formation.

### 2.2. Characterization of the Pd/CS Catalysts

On the HRTEM pictures of the palladium-decorated CS, it is clearly visible that the surface of the carbon foils is richly coated by very small nanoparticles (Figure 3A). The distribution of these nanoparticles on the catalysts surface is continuous. The average particle diameter of the palladium crystallites is 10.1 ± 3.7 nm, which is similar to the 12 ± 3 nm measured by XRD (Figure 3B,C). On the XRD pattern, reflections are found at 40.1°, 46.6°, and 68.1° two-theta degrees, which belong to the Pd(111), Pd(200), and Pd(220) phases, respectively (Figure 3C) (PDF 46-1043). The C(002) reflection is located at 24.1° two-theta degree. 

### 2.3. Results of the Hydrogenation Reactions

Analysis of the reaction mixtures showed that the Pd/CS catalyst was able to aid the hydrogenation of benzophenone; however, the conversions after 240 min of the reaction were different depending on the solvent (Figure 4). Meanwhile, methanol and isopropanol gave good results (97% and 99% conversion, respectively), but THF proved to be less effective (72% conversion). Considering the polarity index of the solvents used, it can be seen that methanol (5.1) is more polar than isopropanol (3.9) and THF (4.0). Since the catalyst support (CSA) has a distinctly polar surface, it is expected to be more dispersible in polar solvents. This higher the dispersibility of the catalyst, the easier the accessibility to reactant molecules that can be allowed by catalytically active centrums (Pd nanoparticles) on the surface, resulting in higher conversion rates in more polar solvents compared to less polar ones. The greater polarity of the solvent and hence the improved dispersibility of the catalyst partially contribute to its enhanced catalytic activity. However, regarding the conversion of benzophenone, similarly high values were measured using methanol and isopropanol. A significant difference can be observed in the selectivity of diphenylmethane in the case of the two alcohols. Diphenylmethane selectivity was the best using isopropanol (99%). However, in methanol and THF, the main product was benzhydrol, with only 13% and 1% diphenylmethane selectivity, respectively. Independently of the solvent, no products other than benzhydrol and diphenylmethane were formed during the reduction, and no intermediates (i.e., partly hydrogenated compounds) or side products could be detected. In our previous work, we compared the Pd/Norit catalyst (non-doped, pure form of carbon) in the hydrogenation of benzophenone with the Pd/BCNT catalyst supported by N-doped carbon nanotubes [7]. The solvent used in that case was THF. As a result, we observed that both diphenylmethane and benzhydrol were produced in significant amounts using the non-doped carbon-supported Pd catalyst. In contrast, the nitrogen-doped carbon nanotube-supported Pd catalyst showed higher selectivity towards the formation of benzhydrol. In our present work, we also experienced a similar situation with the nitrogen-doped carbon-supported Pd catalyst, which exhibited full benzhydrol selectivity during the reaction in THF solvent. These results suggest that the incorporated nitrogen atoms change the electron distribution on the catalyst surface to such an extent that it affects the selectivity.

Reusability of the catalyst was also tested. After isolation from the reaction mixture by decantation, the catalyst was washed with methanol and acetone and dried. No reactivation was performed. In the case of the third usage, the catalyst produces only a small decrease in the benzophenone conversion (from 99 to 98%) and diphenylmethane selectivity (from 99 to 96%).

The kinetic curves of the reactions are shown in Figure 5. It can be seen that in methanol (Figure 5B), the benzhydrol concentration after its maximum at about 180 min of reaction time started to decrease, as it was converted to diphenylmethane at an increasing reaction rate. In isopropanol, the highest benzhydrol concentration was measured at approximately 50 min (Figure 5C), after which it converted almost completely to diphenylmethane at the end of the studied time period (240 min). In THF, a reaction time of 240 min was not enough to reach a benzophenone conversion higher than 72%, with practically no diphenylmethane formation at all (Figure 5A).

We compared our catalytic results with other published results and found that all the catalysts tested in benzophenone hydrogenation in the literature were more selective to benzhydrol than diphenylmethane [8,10,11,35,36] (Table 1). Despite the fact that all these catalysts are selective for benzhydrol, we could prepare a catalyst that can be highly selective for diphenylmethane, depending on the solvent. This result could originate from the fact that instead of the commonly used activated carbons or metal–oxides, we used N-doped carbon support. Furthermore, by changing the solvent, this type of support can be fine-tuned in order to achieve either benzhydrol or diphenlymethane selectivity. In THF, the catalyst is selective towards benzhydrol, but in isopropanol, practically only diphenylmethane is produced. These results allow an alternative method for diphenylmethane production for perfume, dye, and cosmetics industries.

## 3. Materials and Methods

### 3.1. Materials

4-Nitroaniline (C_6_H_6_N_2_O_2_) (ThermoFisher GmbH, Kandel, Germany) and 95 wt% sulfuric acid (H_2_SO_4_) (VWR International Ltd., Rue Carnot, Fontenay-sous-Bois, France) were used for the synthesis of the carbon-based catalyst support. Palladium(II) nitrate dihydrate (Pd(NO_3_)_2_ × 2H_2_O) (Alfa Aesar Ltd., Ward Hill, MA, USA) was used to deposit Pd nanoparticles onto the carbon support. Nitrogen (purity 4.0, Messer) and hydrogen (purity 4.0, Messer) were used during the experiments. Benzophenone (C_13_H_10_O) (ThermoFisher GmbH, Kandel, Germany) was used as a reactant during the catalytic hydrogenation tests. 

### 3.2. Characterization Techniques

The structure of the carbon foam was studied by scanning electron microscope (SEM) using a Helios G4 PFIB CXe (Thermo Scientific, Waltham, MA, USA) instrument. High-resolution transmission electron microscopy (HRTEM, Talos F200X G2, Thermo Scientific, Waltham, MA, USA) electron microscope with field emission electron gun, X-FEG, accelerating voltage: 20–200 kV) was used for the characterization of the particle size and morphology in case of the “carbon snake”-supported (CSS) palladium nanoparticles. For imaging and electron diffraction, a SmartCam digital search camera (Ceta 16 Mpixel, 4k × 4k CMOS camera, Thermo Scientific, Waltham, MA, USA) and high-angle annular dark-field (HAADF) detector were used. Sample preparation was carried out from the aqueous dispersion of the nanoparticles by dropping on 300 mesh copper grids (Ted Pella Inc., Redding, CA, USA). For the phase analysis of the noble metal nanoparticles on the support, X-ray diffraction (XRD) was applied. Bruker D8 diffractometer (Cu-Kα source) in parallel beam geometry (Göbel mirror) with Vantec detector was used. The average crystallite size was calculated by the mean column length calibrated method by using full width at half maximum (FWHM) and the width of the Lorentzian component of the fitted profiles. The zeta potential of the carbon support was measured in aqueous phase by determining the electrophoretic mobility of the particles (laser Doppler electrophoresis) using a Malvern Zetasizer Nano ZS (Malvern Panalytical, Malvern, UK). The palladium content of the catalyst was measured by a Varian 720 ES (Varian Medical Systems, Palo Alto, CA, USA) inductively coupled optical emission spectrometer (ICP-OES). The specific surface area (SSA) analysis was carried out by using N_2_ adsorption–desorption method at 77.3 K. For this purpose, a Micromeritics ASAP 2020 (Micromeretics, Norcross, Georgia, United States) equipment was used. The evaluation was based on the Brunauer–Emmett–Teller (BET) isotherm. For the identification of the chemical bonds of nitrogen and the type of the oxygen-containing functional groups, X-ray photoelectron spectroscopy (XPS) was applied. During the measurements, a Kratos XSAM-800 XPS (Kratos Analytical Ltd., Manchester, UK) instrument was used with a MgKα X-ray source operated at 120 W (12 kV, 10 mA). 

Gas chromatography coupled with mass spectrometry (GC-MS) is a suitable method for both qualitative and quantitative determination of the possible reaction products. The progress of hydrogenation was followed by a Shimadzu GCMS-QP2020 (Shimadzu, Kyoto, Japan) mass spectrometer–gas chromatograph. The reaction mixture contained 1-decanol (830 mgL^–1^) as an internal standard. Chromatographic separation was accomplished using a Stabilwax-MS capillary column (30 m length × 0.25 mm i.d., 0.25 µm film thickness, Bellefonte, PA, USA) from Restek Corp. The column temperature program was as follows: 403 K (1 min), 403–523 K (10 K min^–1^), 523 K (2 min). The injector and the detector were set at 523 K and 503 K, respectively. Helium was used as carrier gas at a 0.86 mL min^−1^ column flow rate. Electron ionization (EI) mode was performed with 70 eV electron energy. The MS cut time was 3 min. A 1 μL sample from the reaction mixture was injected by a Shimadzu AOC-6000 autosampler (Shimadzu, Bellefonte, PA, USA). The split was set at 1:300. 

### 3.3. Preparation of the “Carbon Snake” Carbon-Foil

4-Nitroaniline (1.50 g) in powdered form and 1 mL concentrated (95 wt%) sulfuric acid were mixed in a ceramic crucible. The mixture was heated using a Bunsen burner to effect carbonization. The slowly forming carbon was quickly blown up by the formed gases (CO_2_ and NH_3_) from the thermal decomposition of 4-nitroaniline. The structure of the carbon form, the so-called “carbon snake” (CS), is similar to the conventional foams, but in this case, the foam membranes are made of carbon layers (Figure 6). The CS samples were washed five times with distilled water and dried at 378 K overnight. The activation was carried out in two steps. First, the sample was heated at 673 K for 30 min under nitrogen flow, then it was heated at 1173 K for another 30 min in carbon dioxide atmosphere. 

### 3.4. Deposition of the Catalytically Active Palladium Metal Nanoparticles onto the Surface of the Carbon Nanofoils of CS

The carbon support (2.00 g) was dispersed in 100 mL distilled water by ultrasonication using a Hielscher UIP1000 hDT homogenizer. Into the carbon foil dispersion was added 50 mL aqueous solution of 0.250 g palladium(II) nitrate, and it was treated by ultrasonication for 10 min. The liquid phase was eliminated by a rotary vacuum evaporator, then the samples were dried at 378 K overnight. The precious-metal-ion-containing carbon foils were finally activated in hydrogen atmosphere at 673 K. During this process, the palladium ions were reduced to catalytically active metal nanoparticles. The theoretical metal content of the catalysts is 5 wt%. ICP-OES measurements showed 3.96 wt% palladium content.

### 3.5. Catalytic Tests of the Carbon-Foil-Supported Catalyst in Benzophenone Hydrogenation

The hydrogenation of benzophenone was carried out in isopropyl alcohol (IPA). The concentration of benzophenone was 0.010 mol L^–1^. Then, 0.10 g catalyst was added to 150 mL benzophenone solution. A Büchi Uster Picoclave reactor system equipped with a 200 mL stainless-steel vessel and a heating jacket was used for the hydrogenation tests. The pressure of H_2_ was kept at 20 bar, and the reaction mixture was thermostated at 323 K. Sampling was carried out after 5, 10, 15, 20, 30, 40, 60, 80, 120, 180, and 240 min. 

The efficiency of the catalyst was characterized by calculating the conversion (X%) of benzophenone based on the following Equation (1).
(1)$$X\;\%=\frac{\;\;n_{consumed\;benzophenone}}{n_{initial\;benzophenone}}\;\cdot \;100$$

Diphenylmethane (DPM) yield (Y%) was also calculated as follows in Equation (2).
(2)$$Y\;\%=\frac{n_{formed\;DPM}}{n_{\;theoritical\;DPM\;}}\;\cdot \;100$$

Furthermore, diphenymethane (DPM) selectivity (S%) was calculated according to the following Equation (3).
(3)$$S\;\%=\frac{n_{formed\;DPM}}{{\sum n}_{\;products\;}}\;\cdot \;100$$

## 4. Conclusions

In this research, a palladium0containing carbon-supported catalyst was prepared, characterized, and tested in ketone hydrogenation. At 323 K and 20 bar in isopropanol, the catalyst showed good activity in benzophenone reduction to diphenylmethane, with 99% conversion and 99% selectivity. The catalyst can be simply reused at least two more times without any complex reactivation process, with only 1–3% loss in conversion and selectivity. In other solvents (methanol and THF), lower conversion and selectivity were achieved. The “carbon snake”-supported catalyst proved to be more efficient in diphenylmethane production than the commercial Norit-based palladium catalyst, as tested in our previous work [7].

## Figures and Tables

**Figure 1 ijms-24-12211-f001:**
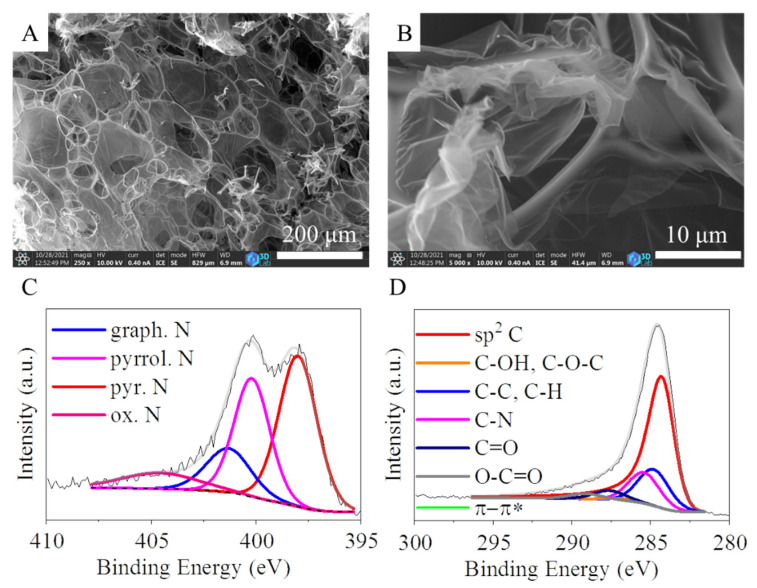
SEM pictures (**A**,**B**) and the deconvoluted N 1s band (**C**) and C 1s band (**D**) on the XPS spectrum of the carbon support.

**Figure 2 ijms-24-12211-f002:**
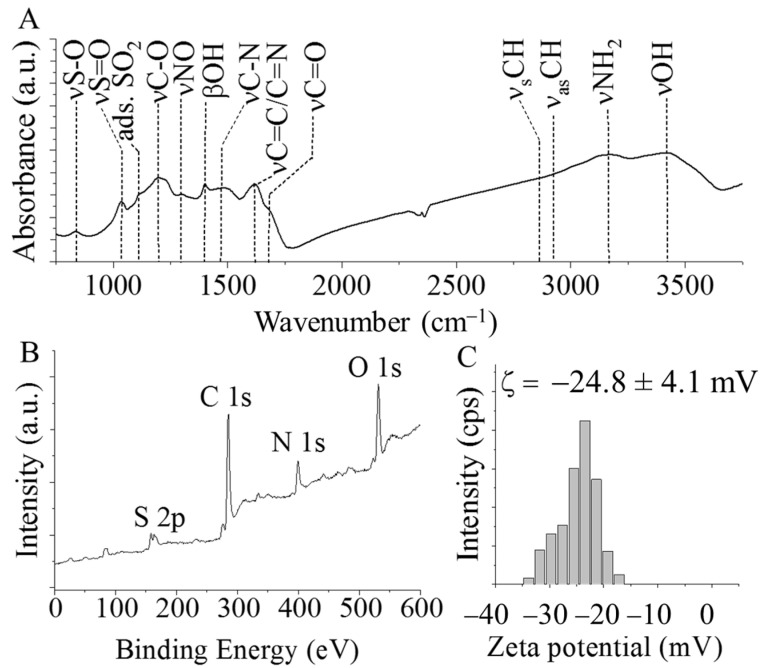
FTIR (**A**) and XPS spectrum (**B**) and zeta potential distribution (**C**) of the CS catalyst support.

**Figure 3 ijms-24-12211-f003:**
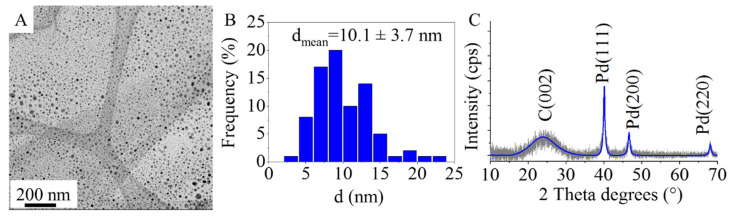
TEM picture (**A**), size distribution histogram (**B**), and XRD pattern (**C**) of the Pd/CS catalyst.

**Figure 4 ijms-24-12211-f004:**
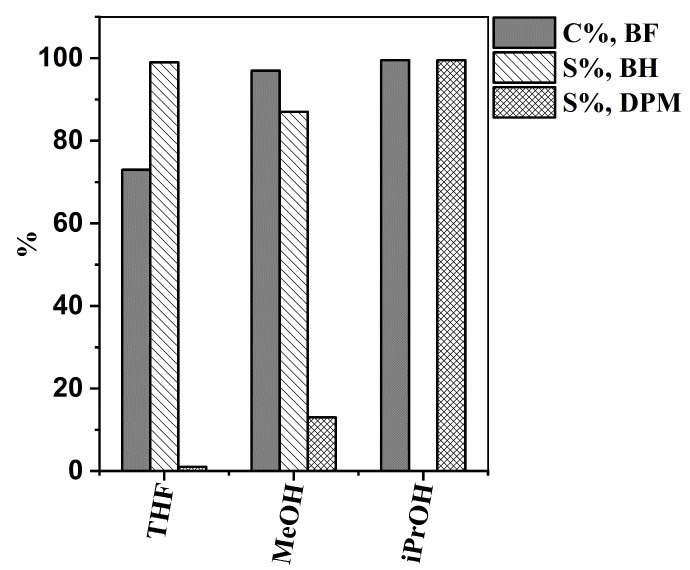
Benzophenone conversions (C%, BF), benzhydrol selectivities (S%, BH), and diphenylmethane selectivities (S%, DPM) in different solvents.

**Figure 5 ijms-24-12211-f005:**
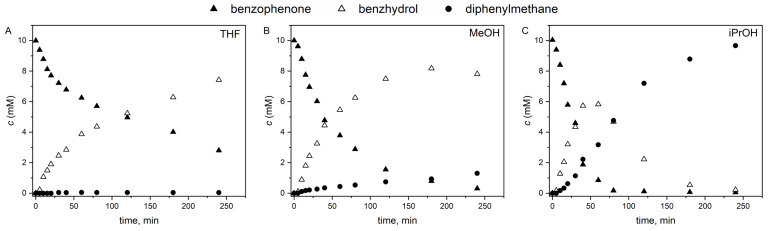
Kinetics of catalytic benzophenone hydrogenation using Pd/CS catalyst in (**A**) THF, (**B**) methanol, and (**C**) isopropanol solvent.

**Figure 6 ijms-24-12211-f006:**
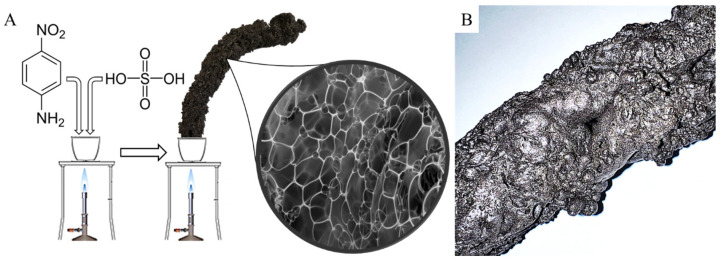
Synthesis of the “carbon snake” (CS) by carbonization of 4-nitroaniline (**A**) and photo of the produced foam-like carbon form (**B**).

**Table 1 ijms-24-12211-t001:** The comparison of the Pd/Cs catalyst in benzophenone hydrogenation with others in the literature.

Catalyst	Solvent	T (K)	p (bar)	BF Conv. (%)	BH SELECT. (%)	DPM Select. (%)	Ref.
Pd/CS	THF	323	20	72	99	1	This work
Pd/CS	Isopropanol	323	20	99	-	99	This work
Pd/CS	Methanol	323	20	97	87	13	This work
Ni-Fe/TiO_2_	Methanol–water–NaOH	408	60	88	98.4	N.A.	[8]
Ni-Fe/TiO_2_	Methanol	408	60	86.7	84.5	13.9	[8]
Ni-Fe/TiO_2_	Isopropanol	408	60	86	80	N.A.	[8]
Ni-Fe/TiO_2_	Cyclohexane	408	60	84	58.8	N.A.	[8]
Pd/act. C	Isopropanol	323	15	93	100	-	[10]
Ni-La-B	Methanol	403	25	99.8	90	6.9	[11]
Raney Ni	Methanol	403	25	31.3	78.1	1	[11]

## Data Availability

Data sharing not applicable.

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
