# Peer review of "Preparation and Testing of a Palladium-Decorated Nitrogen-Doped Carbon Foam Catalyst for the Hydrogenation of Benzophenone"

_ijms, 2023, doi:10.3390/ijms241512211_

Round 1

Reviewer 1 Report

The manuscript is readable and well-arranged. In general, has potent to be highly cited, but it cannot be accepted in its present form. Some errors and organizational issues need to be addressed before meriting publication. Please consider the following comments:

general questions:

- Zeta potential of an insoluble, hardly dispersible material? How and why?

- SSA with CO2 for functionalized material? Why not N2?

- Sorry, but there is still a long way from CS to graphene!!!

Detailed issues:

- Based on your XPS results: N - 13.4 %at. means you CANNOT use CO2 for SSA measurements!!!!

- Nice pictures, but it doesn't mean that you have a structure similar to graphene, if you insist you have to prove it!!!

- I'm sorry, but from your FTIR spectrum, it can be interpreted that your sample contains a lot of sulfur functional groups, which makes sense as your CS was prepared in H2SO4. To prove me wrong, you need to show at least the entire XPS spectrum and EDX

- More on the IR interpretation of carbon materials would be nice to discuss (e.g. 10.1016/j.cattod.2006.08.037 or other works from this group)

- Additionally (i) it would be better without baseline correction; (ii) the attribution is a bit wrong

- The statement that carbons are not good materials for measuring zeta potential is supported by an extremely broad signal.

- I'm sorry, but comparing about 100% with about 100% makes no sense in my opinion. Please remove Fig.5 as all you need is in Fig. 6

The discussion on p. 8, ll 275-288 is nice, but perhaps a table would also be valuable

The authors should recheck the paper for grammar errors and misleading sentences. 

Reviewer 2 Report

The manuscript entitled as "Preparation and testing of a palladium decorated nitrogen-doped carbon foam catalyst for the hydrogenation of benzophenone" is an interesting work presenting a detailed research for a catalyst design for the hydrogenation of benzophenone. Some minor questions need to be answered by the authors.

1. In the Introduction section there is not that much information concerning the catalysts which have been applied for the presented reaction and their corresponding results, so as the reader to be in position to compare the results with corresponding literature findings at the end of the manuscript.

2. In p. 5, lines 13-174, it is referred that "nitrogen doping of carbon can also influence the selectivity of the catalyst". The explanation which is provided in the following lines justifies the effect on the conversion. The influence on selectivity is not straightforward. Can the authors provide an explanation?

3. In Section 3.3, the results of the catalytic testing are described. Can the authors explain or do they have an idea if the observed results are due to the catalyst, due to the support or due to the solvent? It is clear that there is an all in all combinatorial effect of the above-mentioned three factors, but if for example the catalyst is not doped and the solvent will change, will the results be the same as those presented in the manuscript?

4. In Section 3.3, p. 7, lines 250-251, it is mentioned that in MeOH and THF the main product was benzhydrol and the selectivity to diphenylmethane was very low. Do the authors have an explanation for this.

Only few typos, such as some commas and full stops missing from manuscript.

Round 2

Reviewer 1 Report

the manuscript has been sufficiently improved, is worth publication in IJMS